# Properties of Therapeutic Deep Eutectic Solvents of l-Arginine and Ethambutol for Tuberculosis Treatment

**DOI:** 10.3390/molecules24010055

**Published:** 2018-12-24

**Authors:** Filipa Santos, Maria Inês P.S. Leitão, Ana Rita C. Duarte

**Affiliations:** 1LAQV, REQUIMTE, Departamento de Química da Faculdade de Ciências e Tecnologia, Universidade Nova de Lisboa, 2829-516 Caparica, Portugal; mfca.santos@campus.fct.unl.pt; 2ITQB–Instituto de Tecnologia Química e Biológica, Universidade Nova de lisboa, Estação Agronómica Nacional, Av. Da República, 2780-157 Oeiras, Portugal; inesleitao@itqb.unl.pt

**Keywords:** therapeutic deep eutectic solvents, tuberculosis, ethambutol, l-arginine

## Abstract

The treatment for tuberculosis infection usually involves a prolonged regimen of multiple antibacterial drugs, which might lead to various secondary effects. For preventing drug resistance and side-effects of anti-tuberculosis drugs, new methods for improving the bioavailability of APIs were investigated. The strategy proposed consists of the preparation of therapeutic deep eutectic solvents (THEDES), that incorporate l-arginine and ethambutol. The eutectic mixtures were prepared by mixing the components at a certain molar ratio, until a clear liquid solution was formed. The prepared mixtures were characterized by differential scanning calorimetry (DSC), polarized optical microscopy (POM) and nuclear magnetic resonance spectroscopy (^1^H and ^13^C-NMR). The solubility and permeability of the drugs when they are in the THEDES form was evaluated at 37 °C, in phosphate buffered saline (PBS). Solubility studies showed an increase of the solubility of ethambutol when incorporated in the eutectic system. The cytotoxicity was evaluated using a model cell line (Caco-2), comparing the cytotoxicity of the API incorporated in the eutectic system. We observed that the cell viability in the THEDES was affected by the presence of citric acid, and higher cytotoxicity values were observed. Nonetheless, these findings do not compromise the possibility to use these systems as new delivery systems for ethambutol and arginine.

## 1. Introduction

Tuberculosis infection caused by *Mycobacterium tuberculosis complex*, continues to be one of the major causes of morbidity and mortality worldwide, and the treatment frequently comprises a combination of multiple antibacterial drugs administered for several months [1,2,3]. The World Health Organization (WHO) estimated 10.4 million of new cases of tuberculosis, in 2016, and 1.7 million died from the disease, including HIV-positive patients [1]. Considering data reported to WHO and the estimations of new cases of drug-resistance to first-line drugs, like rifampicin, isoniazid, ethambutol, pyrazinamide or rifabutin becomes necessary develop improvements for anti-tuberculosis drugs [4]. The WHO establishes milestones to reduce the mortality caused by tuberculosis infections and accelerate the decline of the infection, particularly when multidrug-resistance exists. Also, the increase of highly resistant strains of tuberculosis makes it necessary for development of new drugs for treatment of this epidemic disease [4].

In recent years, green chemistry approaches have been considered for the development of new drugs or improvement of their properties while at the same time help in the implementation of more sustainable processes for their production. The preparation of therapeutic deep eutectic solvents (THEDES) by incorporating the active pharmaceutical ingredient (API) in the system or solubilizing the API in a previous synthesized eutectic mixture [5], is a sustainable preparation method that could be used to improve bioavailability of the drugs. A eutectic mixture is defined as a combination of two or more components in a certain molar ratio, that by hydrogen bonding form a new compound characterized by a melting point depression, compared with the melting temperatures of the initial compounds [5,6,7,8,9].

Currently available therapies for tuberculosis infections involve the combination of four anti-tuberculosis drugs, namely isoniazid, rifampicin, ethambutol and pyrazinamide in the first months and then the treatment continues with isoniazid and rifampicin [2,3,4,10]. Ethambutol is an anti-tuberculosis drug that is bacteriostatic (inhibits the cell wall synthesis of bacteria) and helps to prevent emergence of rifampicin resistance, when resistance to isoniazid is present [3,11]. Arginine is a basic and semi-essential amino acid that is naturally ingested in diet. It was reported that arginine reduces the symptoms associated to tuberculosis, in patients with tuberculosis and without HIV infection. The supplementation with l-arginine could improve the immune system by increasing the concentration of nitric oxide that acts as a cytotoxic substance in pathological processes. The high levels of nitric oxide produced by arginine in T-helper lymphocytes type 1 (Th1), activate macrophages, which are important cells in the immune response to tuberculosis infection. Studies demonstrate that arginine could be useful as an adjunctive therapy for patients with tuberculosis and those, who are mediated by the increase production of nitric oxide [12,13,14].

In this study, new THEDES based on ethambutol and l-arginine were prepared to enhance the properties of these substances, particularly to increase their solubility, permeability and hence, in the future, be able to lower any concentrations in the prescribed treatments.

## 2. Results

### 2.1. Polarized Optical Microscopy

The polarized optical microscopy (POM) was studied to observe the systems under polarized light with a cross polarizer and to detect the existence of different phases in the eutectic mixtures. We could observe black images in all samples (Figure 1), as well as characteristics of an amorphous and homogeneous liquids without formation of crystals.

### 2.2. Differential Scanning Calorimetry

As we can observe in Figure 2, the thermogram of the initial compounds present peaks of endothermic events, in some of them we observe two peaks which could indicate different transition stages between the crystalline and amorphous phases [6,8,15,16,17]. THEDES with ethambutol in liquid phase does not present peaks at endothermic stage. In THEDES with l-arginine, a peak close to 100 °C which is mostly due to water evaporation at this temperature, is observed.

The thermal behavior of these new compounds is characterized for THEDES with ethambutol, which shows absence of peaks, as well as the formation of small peaks for THEDES with l-arginine, which is due to the water evaporation. Consequently, the eutectic mixtures were successfully formed and demonstrate the existence of interactions between the compounds that form THEDES by hydrogen bonding [6].

### 2.3. Nuclear Magnetic Resonance Studies

NMR spectroscopy was performed and ^1^H and ^13^C-NMR spectra (Appendix A), HMBC and NOESY in ethambutol and THEDES with ethambutol was obtained, to explore possible interactions between the molecules that form THEDES, by hydrogen bonding [18,19]. From ^1^H-NMR differences between chemical shifts of the APIs and the THEDES were observed, mainly in the groups -NH of ethambutol that undergoes an upfield shift of 0.1 ppm from the API to THEDES.

In our study ^1^H-^1^H nuclear Overhauser spectroscopy (NOESY) was used to observe direct intermolecular and intramolecular interactions in THEDES with ethambutol (Figure 3), as this technique identifies spatially close protons. The NOESY spectrum shows interactions between the -OH of the -COOH groups of citric acid and the -OH and -NH groups of ethambutol (signal at 12.34 ppm); at 9.14 ppm it can be noticed that the -NH groups of ethambutol are spatially close to the -OH groups of ethambutol and -OH groups of citric acid. At 5.40 ppm the interaction between the -OH groups of ethambutol -NH group of ethambutol and the -OH groups of citric acid was observed. Furthermore, the spectrum shows an interaction between the -C-OH group of citric acid and the groups -OH and -NH of ethambutol (signal at 5.15 ppm). Finally, strong interactions between these groups and water is also shown.

### 2.4. Solubility and Permeability Studies

The study of solubility and permeability of the API and THEDES was evaluated in PBS, at 37 °C to simulate physiological conditions. The results obtained are presented in the Table 1 and Figure 4.

We can observe an increase of the solubility of ethambutol in THEDES form of approximately 27.5 fold (from 4.64 mg/mL to 127.60 mg/mL), compared with ethambutol. However, in the case of systems with l-arginine only a slight increase of solubility was observed compared to the pure API.

The permeability and diffusion coefficients were studied for the APIs and THEDES which presented enhanced performance in the solubility studies and have some variability in molar ratios of citric acid and water, with higher ratios of water. The assays were carried out using a commercially available polyethersulfone (PES-U) membrane. A rapid diffusion rate was observed in the first minutes and then the diffusion rate stabilizes. The permeability and the diffusion coefficient through the membrane was estimated according to equations 1 and 2 and presented in Table 2.

In the case of ethambutol the permeability and diffusion coefficient change compared to the THEDES, and the systems present a significant increase on the permeability, however in systems with l-arginine the permeability is lower in THEDES and the diffusion coefficient that represents the amount of compound that is diffused through the membrane did not present significant differences.

### 2.5. Biological Performance

In this work, we have evaluated the cell viability in the presence of the THEDES and the initial compounds using a model Caco-2 cell line, that is originated from human colon epithelial cancer cell line, and when cultured as a monolayer spontaneously acquires characteristics of mature enterocytes, which is completed after 7 days of seeding. This characteristic model when grown to confluence could mimic the human small intestine and might be used to test the intestinal permeability in vitro [20].

We observed through the assay with MTS for cell viability that citric acid is more toxic than ethambutol, which contributes for the toxicity of THEDES systems (Figure 5).

After the MTS assay the IC_50_ in Caco-2 cell line was determined and the pH of the solutions in culture media was measured (Figure 6 and Table 3). It was observed values of IC_50_ higher for ethambutol, l-arginine and THEDES CA:l-Arg:H_2_O (1:1:7). With exception of l-arginine all the pH values of the solutions in culture media RPMI are acidic, since every THEDES have higher acidity comparing with any of the initial compounds, which could contribute to increase the toxicity of some compounds.

## 3. Discussion

The preparation of novel eutectic systems continues to be a challenge because the mechanisms that trigger the hydrogen bonding formation in the systems are not yet completely understood [6]. In this work, the aim was to produce therapeutic deep eutectic systems with anti-tuberculosis APIs incorporated. After optimization, six different eutectic systems (Appendix A) were obtained and clear and transparent liquids at room temperature.

The THEDES were studied with POM and black images were observed, which indicates amorphous and homogeneous liquids, without the formation of any solid crystals. In the thermogram was evaluated by DSC and it well defined phase transitions of individual components were observed. Around 100–120 °C, within THEDES phase transitions appear in the thermogram, which indicate that these systems start to be unstable at these temperatures. This is attributed to the ratio of water that evaporates from the systems at these temperatures. According to literature, the systems must have melting temperatures below the melting peaks of the individual components [6,16]. In binary mixtures these observations lead to the absence of melting peaks of each individual component in DSC diagram [19,21]. The results obtained with DSC and POM indicate that systems are above the respective liquid line on the phase diagram [6]. Dai and co-workers have found that small amounts of water resulted in systems that are liquid at room temperature, reduce the time of preparation and the temperature could decrease their viscosity. The stability of the DESs with small amounts of water, generally is increased and the solubilizing capacity could also be tuned. However, adding superior amounts of water (up to 50% of water content) could lead to dilution of DESs and results in a loss of the existing hydrogen bonding, characteristic of these solvents [16,18,21]. In the case of the THEDES herein reported, the presence of water is fundamental to the preparation of the systems.

With NMR studies we were able to study the structure of the compounds and confirm the presence of citric acid and ethambutol in THEDES, by ^1^H and ^13^C-NMR, HMBC and NOESY. Interactions between the compounds, particularly in the amine groups of ethambutol, were detected. These presented a delocalization of the chemical shift (9.24 ppm to 9.14 ppm), which we hypothesized that is due to interactions by hydrogen bonding formed with -OH of citric acid and water. In which concerns the hydrogen bonded protons, changes in the chemical shift could be observed when THEDES form a liquid structure, as we observed the upfield shift in the amine groups of ethambutol [5].

The solubility of the compounds is an important parameter for evaluating the features of the THEDES prepared and observe if they are promising for future incorporation on medicines. In our study was observed a significant increase of the solubility of ethambutol in the THEDES, however for the systems containing l-arginine the difference was not so noticeable. In the case of the permeability studies of the different compounds, similar behavior was observed. While the permeability of ethambutol was increased in THEDES form, in case of arginine the THEDES presented a lower permeability compared to the pure API. The diffusion coefficient of the systems presented the same characteristic behavior of permeability, as higher diffusion coefficients means faster diffusion of the compound through the membrane.

Duarte et al. described that a higher solubility leads to higher driving force through the membrane in permeability studies [5], and our results also confirm this. As expected, the permeability and diffusion coefficient of ethambutol in THEDES was higher, as its solubility is highly increased. This was not the case in the systems with l-arginine.

The results obtained in solubility and permeability studies permit fit the THEDES in the biopharmaceutics classification system and observe an increase of the properties of solubility and permeability in THEDES with ethambutol. The parameters of biopharmaceutics classification system are described in literature, and classifies the APIs in four different classes, since class I is high soluble and high permeable; class II is low soluble and high permeable; class III is high soluble and low permeable, and class IV is low soluble and low permeable [5]. An API is considered highly soluble when the volume for dissolve 1 mg is lower than 250 mL, which we could verify in THEDES with ethambutol that present a solubility of 127.6 mg/mL. A compound is considered highly permeable if presents permeability higher than 6 × 10^−6^ cm s^−1^ and the THEDES with ethambutol presented a permeability of 128.3 × 10^−5^ cm s^−1^ that is higher than the reference value. According to literature and the Biopharmaceutical Classification System (BCS), ethambutol is classified as a class III compound, i.e., a compound with high solubility but low permeability. The results obtained in this work show that THEDES can tailor the bioavailability of API’s and we can postulate that ethambutol can change its classification from class III to class I, being in the THEDES form, highly soluble and highly permeable [5,22].

The biological performance of the systems has shown a higher cytotoxicity of the THEDES in respect to the pure API’s which is due to the presence of citric acid. In any case, as these experiments demonstrate a dose-response they continue to be promising for anti-tuberculosis treatment as it is possible to adjust the dosage required and decrease the dose due to the higher solubility and permeability observed, particularly for the system containing ethambutol.

## 4. Materials and Methods

### 4.1. Materials

The reagents used for the preparation of THEDES were citric acid monohydrate (CA), (CAS number 5949-29-1, ≥99.5% purity) obtained from PanReac AppliChem (Barcelona, Spain); l-arginine (l-Arg), (CAS number 74-79-3, ≥98% purity) obtained from Sigma Aldrich (St. Louis, MO, USA); and ethambutol (ETH), (CAS number 74-55-5, ≥98% purity) was obtained from Alfa Aesar (Haverhill; MA, USA). The compounds were used without additional purification. Phosphate buffer saline (PBS) were prepared as indicated in tablets of PBS by Fisher BioReagents (Hampton, NH, USA), one tablet dissolved in 200 mL of deionized water, yielding a 0.01M phosphate buffer, 0.0027 M potassium chloride, 0.137 M sodium chloride, pH 7.4 solution, at 25 °C.

### 4.2. Preparation of THEDES

Various systems with ethambutol and l-arginine in different combinations of molar ratios were prepared (Appendix A), however the mixtures indicated in the Table 4 were the ones that we observe to form a stable eutectic mixture. The different components of the eutectic mixtures were weighed and mixed, accordingly to different molar ratios, presented in Table 4. The THEDES were obtained by heating the combination until 50/60 °C, under constant stirring (300 rpm) between 6 to 10 h, until a clear liquid solution was formed [5,6].

### 4.3. Differential Scanning Calorimetry (DSC) Analysis—Thermal properties

The DSC experiments were performed using a DSC 131 instrument (Setaram, Tokyo, Japan) operating with temperatures between −150 and +550 °C. The measurements were performed under dry nitrogen atmosphere (at a flow rate of 50 mL min^−1^), with samples of 5–20 mg packed in hermetic aluminium pans. The samples of initial compounds were equilibrated at 20 °C for 5 min, from 5 °C up to 230 °C for ethambutol; for l-arginine up to 250 °C and for citric acid up to 170 °C; followed by an isothermal period of 2 min, and cooling to the assay end temperature at a cooling rate of 10 °C min^−1^. The samples with THEDES were equilibrated at 20 °C for 5 min, from 5 °C up to 230 °C for ethambutol systems and up to 260 °C for systems with l-arginine, followed by an isothermal period of 2 min, and cooling to the assay end temperature at a cooling rate of 10 °C min^−1^.

### 4.4. Polarized Optical Microscopy (POM)

Optical characterization of the THEDES were carried out at room temperature using a transmission mode of an BX-51 polarized optical microscope (Olympus, Tokyo, Japan) connected to an Olympus KL2500 LCD cold light source. A droplet of THEDES was placed on a microscope glass slide and then observed. The images were obtained with an equipped camera (Olympus DP73) and Olympus Stream Basic 1.9 software (Olympus, Tokyo, Japan).

### 4.5. Nuclear Magnetic Resonance Studies (NMR)

All NMR spectra ^1^H and ^13^C were recorded at room temperature on a Bruker (Billerica, MA, USA) spectrometer (400 MHz) and chemical shifts were referenced to SiMe_4_ (δ in ppm). The THEDES samples were placed in a 5 mm NMR tube with deuterium dimethyl sulfoxide (DMSO-d_6_).

### 4.6. Solubility and Permeability Studies

The solubility studies were made with the pure APIs and THEDES. For performing these assays were used a PBS solution, and was added an excess of API and THEDES to this solution, place at 37 °C, under stirring for 24 h, for simulate physiological conditions. The determination of the API solubility was made by UV spectroscopy in 1 mL quartz cuvettes, using a Lambda 35 UV/VIS spectrometer (PerkinElmer, Villepinte, France) and the PerkinElmer UV WinLab software (PerkinElmer, Villepinte, France). The absorbance of the solutions was measured at the maximum wavelength of the respective API, 203 nm (ETH) and 211 nm (l-Arg). The calibration curve was made using the respective APIs as standard for quantification.

For the permeability studies was used a glass Franz diffusion cells (PermeGear, Hellertown, PA, USA) with 8 mL in the receptor compartment, 2 mL in the donor compartment and an effective mass transfer area of 1 cm^2^. The membrane used was a polyethersulfone (PES-U) membrane, with 150 μm thickness and 0.45 μm pore size (Sartorius Stedim Biotech, Goettingen, Germany). The membrane was place between the receptor and donor compartment and held with a stainless-steel clamp. The receptor compartment was filled with a PBS solution and the air bubbles were removed by carefully tilting the Franz cells for the air bubbles to escape through the sampling arm. Then the donor compartment was filled with the API or THEDES and 2 mL of PBS solution. The samples were taken in intervals of 3, 5, 10, 15, 20, 25, 30, 40, 50, 60, 90, 120, 150 and 180 min, and the amount of diffused API were measured by UV spectroscopy in 1 mL quartz cuvettes, using the Lambda 35 UV/VIS spectrometer and the PerkinElmer UV WinLab software. The experiments were performed at 37 °C in a water bath with stirring at 60 rpm, to eliminate the boundary layer effect. The permeability (*P*) of APIs and THEDES was calculated by the following equation (Equation (1)):(1)$$-\ln\left(1-\frac{2C_{t}}{C_{0}}\right)=\frac{2A}{V}\times P\times t$$
where *C*_t_ is the concentration in the receptor compartment at time *t*, *C*_0_ is the initial concentration in the donor compartment, *V* is the volume in two compartments, and *A* is the effective area of permeation. The permeability coefficient can be calculated from the slope of the curve −(*V*/2*A*) * ln(1 -2*C*_t_/C_0_) versus *t* (cm s^−1^) [5,24,25].

The diffusion coefficient (*D*) of APIs and THEDES across the membrane was calculated according to Fick’s Law of diffusion, following the equation (Equation (2)):(2)$$D=\frac{V_{1}V_{2}}{V_{1}+V_{2}}\times \frac{h}{A}\times \frac{1}{t}\ln\left(\frac{C_{f}-C_{i}}{C_{f}-C_{t}}\right)$$
where *D* is the diffusion coefficient (cm^2^ s^−1^), *C*_i_ and *C*_f_ are the initial and final concentrations, and *C*_t_ is the concentration at time *t* in the receptor compartment (mol L^−1^), *V*_1_ and *V*_2_ are the volume of liquid in donor and receptor compartment, respectively (cm^3^), *h* is the thickness of the membrane (cm) and *A* is the effective diffusion area of the membrane (cm^2^) [5,24,25].

### 4.7. In Vitro Cytotoxicity and IC_50_ Evaluation

For the evaluation of the biological performance of the THEDES, were made studies with Caco-2 cells, a human colon epithelial cancer cell line used as a model of human intestinal absorption of drugs and other substances. Caco-2 cell line was provided by DSMZ (Braunschweig, Germany, ACC 169) and was cultured in RPMI (1640 medium powder from Gibco, Lisboa, Portugal) supplemented with 10% heat-inactivated fetal bovine serum (FBS) and 1% of antibiotic (penicillin-streptomycin). Cells were maintained at 37 °C in a humidified incubator with 5% of CO_2_ [26]. Cell culture medium and supplements were obtained from Gibco (Life Technologies—Alfagene, Lisboa, Portugal). The cytotoxicity assay in Caco-2 cell line was performed after a culture period of 7 days, for the cells achieve a monolayer of growth with approximately 80–90% confluence. Then the cells were seeded in a 96-well culture plate and incubated with 100 μL of different concentrations of THEDES and initial compounds, for 24 h. After the incubation time the cells were washed with PBS, and 100 μL of MTS (3-(4,5-dimethylthiazol-2-yl)-5-(3-carboxymethoxyphenyl)-2-(4-sulfophenyl)-2*H*-tetrazolium bromide) reagent (1:10 dilution) was added at each well and then incubated for 4 h, at 37 °C in 5% of CO_2_ atmosphere. The amount of formazan product was measured by absorbance at wavelength of 490 nm with and Epoch Microplate Spectrophotometer (Bio-Tek Instruments, Winooski, VT, USA). The data was expressed in percentage of cell viability with the control and the experiments were completed with 3 independent experiments with three replicates.

The IC_50_ evaluation was determined with the percentage of cell viability at different concentrations (between 0.01-1M), with successive dilutions of 1:2 starting in 1M of API or THEDES, using the software GraphPad Prism 6.01 (Graphpad Software Inc., La jolla, San jose, CA, USA) (Appendix A). The measurements of pH solutions were completed with a micropH 2001 of Crison (Barcelona, Spain).

### 4.8. Statistical Analysis

The experiments were made in triplicates for each condition, being the data presented with mean and standard deviation. The statistical analysis was performed using GraphPad Prism 6.01 and following a parametric test (One-Way ANOVA with multiple comparisons to the control) or a *t*-test for pairs of samples. Differences between experimental data were considered significant with a confidence interval of 95%.

## 5. Conclusions

The evaluation and knowledge of these properties in the THEDES synthesized is essential for further studies, namely the evaluation of the antimicrobial activity of theses THEDES and for improving pre-existing drugs bioavailability and hence their administration regimens. This system incorporating the API are a sustainable way to synthesize drugs and represent a progress to improve pharmacokinetics of APIs and could provide new formulations. The systems with ethambutol and CA:l-Arg:H_2_O (1:1:7) seems to be the more promising systems, due to their characteristics presented by water content, solubility and cytotoxicity assays.

## Figures and Tables

**Figure 1 molecules-24-00055-f001:**
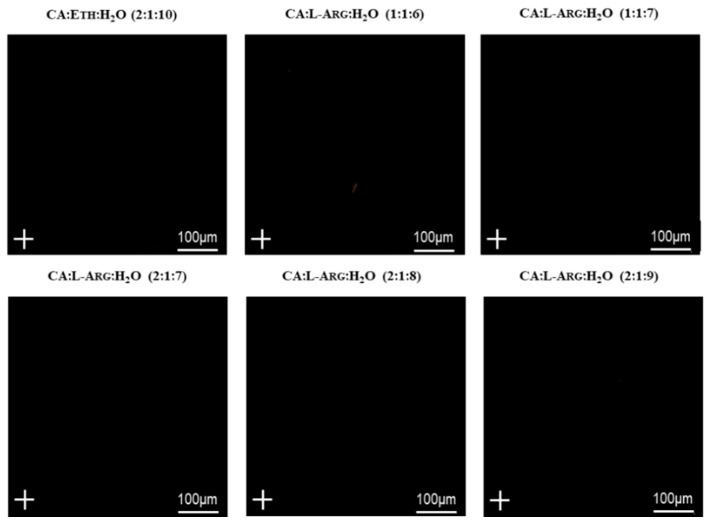
Polarized optical microscopy images of different THEDESs.

**Figure 2 molecules-24-00055-f002:**
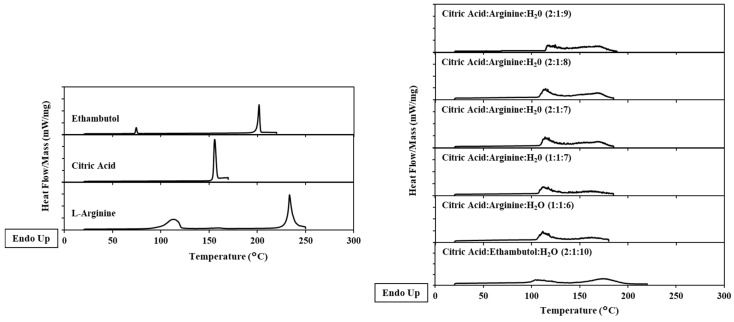
Thermograms obtained in hermetic capsules of initial components (**a**) and THEDES (**b**).

**Figure 3 molecules-24-00055-f003:**
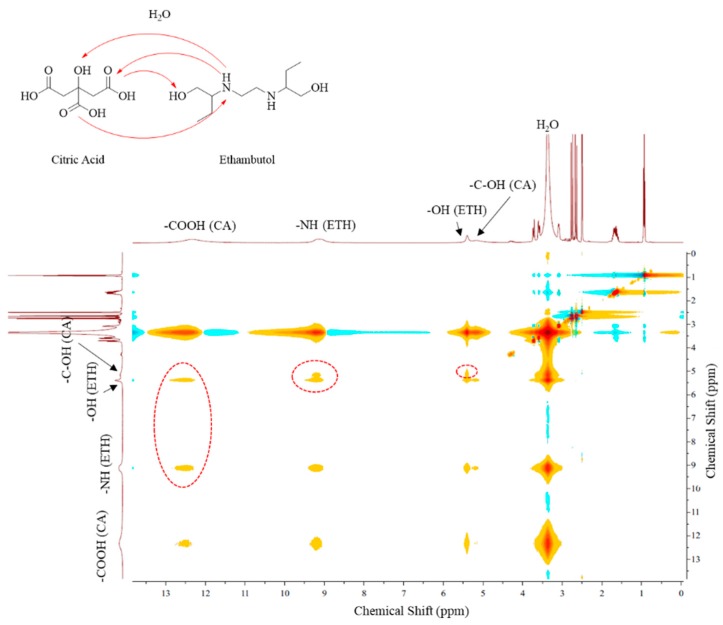
^1^H-^1^H–nuclear Overhauser enhancement spectroscopy (NOESY) with detected interactions between citric acid and ethambutol.

**Figure 4 molecules-24-00055-f004:**
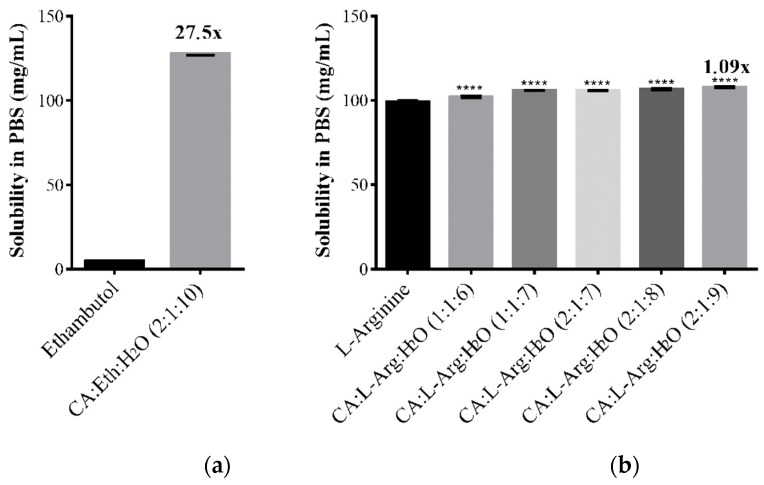
Solubility of THEDES and raw materials with ethambutol (**a**) and l-arginine (**b**), in PBS at 37 °C. Results are presented as mean ± standard deviation and statistically significant differences (****) are shown as *p* < 0.05 in comparison with raw material.

**Figure 5 molecules-24-00055-f005:**
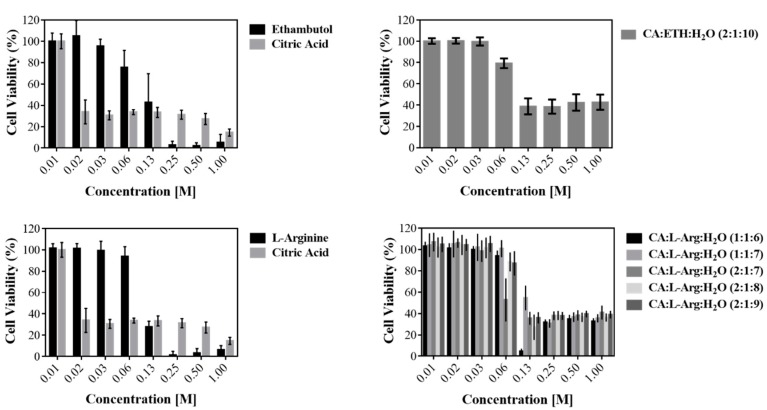
Cell viability (%) measured in Caco-2 cell line, in THEDESs and raw materials.

**Figure 6 molecules-24-00055-f006:**
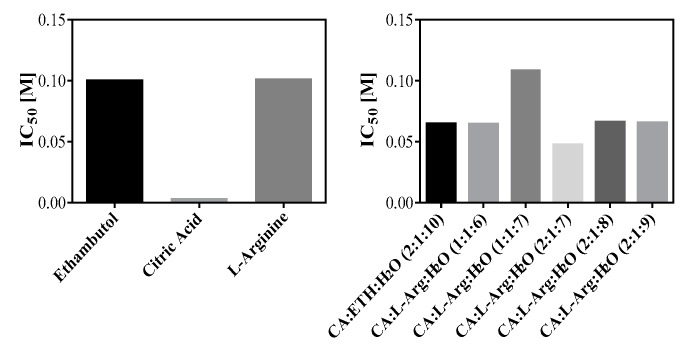
IC_50_ representation, based on cell viability obtained in Caco-2 cell line.

**Table 1 molecules-24-00055-t001:** Solubility of APIs and THEDESs in PBS (pH 7.4), at 37 °C.

THEDES	Molar Ratio	Solubility (mg/mL)
Ethambutol		4.64 ± 0.07
Citric Acid:Ethambutol:H_2_O	2:1:10	127.60 ± 0.69
l-Arginine		99.35 ± 0.54
Citric Acid:l-Arginine:H_2_O	1:1:6	102.17 ± 0.56
Citric Acid:l-Arginine:H_2_O	1:1:7	106.05 ± 0.12
Citric Acid:l-Arginine:H_2_O	2:1:7	105.89 ± 0.15
Citric Acid:l-Arginine:H_2_O	2:1:8	106.77 ± 0.49
Citric Acid:l-Arginine:H_2_O	2:1:9	107.86 ± 0.33

**Table 2 molecules-24-00055-t002:** Permeability and diffusion coefficients calculated for the different systems.

THEDES	Permeability (10^5^ cm s^−1^)	Diffusion Coefficient (10^6^ cm^2^ s^−1^)
thambutol	81.9 ± 3.06	13.6 ± 1.01
Citric Acid:Ethambutol:H_2_O (2:1:10)	128.3 ± 10.4	17.4 ± 1.28
l-Arginine	36.6 ± 2.39	8.20 ± 0.39
Citric Acid:l-Arginine:H_2_O (1:1:7)	21.6 ± 0.21	5.59 ± 0.09
Citric Acid:l-Arginine:H_2_O (2:1:9)	23.2 ± 0.3	5.58 ± 0.1

**Table 3 molecules-24-00055-t003:** IC_50_ of THEDES and raw materials in Caco-2 cell line.

THEDES	Molar Ratio	IC_50_ (M)	pH
Ethambutol		0.1007	6.21
Citric Acid:Ethambutol:H_2_O	2:1:10	0.0647	5.61
Citric Acid		0.0033	7.08
l-Arginine		0.1017	10.43
Citric Acid:l-Arginine:H_2_O	1:1:6	0.0644	6.61
Citric Acid:l-Arginine:H_2_O	1:1:7	0.1085	5.88
Citric Acid:l-Arginine:H_2_O	2:1:7	0.0476	5.79
Citric Acid:l-Arginine:H_2_O	2:1:8	0.0662	5.48
Citric Acid:l-Arginine:H_2_O	2:1:9	0.0656	5.79

**Table 4 molecules-24-00055-t004:** Different THEDES prepared.

Component A	Component B	Component C	Molar Ratio	Molar Mass * (g/mol)	Abbreviation	Visual Aspect
Citric Acid	Ethambutol	H_2_O	2:1:10	61.02	CA:ETH:H_2_O	Transparent liquid at room temperature
Citric Acid	l-Arginine	H_2_O	1:1:6	61.56	CA:L-Arg:H_2_O	
Citric Acid	l-Arginine	H_2_O	1:1:7	56.72	CA:L-Arg:H_2_O	
Citric Acid	l-Arginine	H_2_O	2:1:7	77.02	CA:L-Arg:H_2_O	
Citric Acid	l-Arginine	H_2_O	2:1:8	67.15	CA:L-Arg:H_2_O	
Citric Acid	l-Arginine	H_2_O	2:1:9	63.06	CA:L-Arg:H_2_O	

* M_DES_ = X_salt_ × M_salt_ + X_HBD_ × M_HBD_, where X and M are mole fraction and molar mass, respectively [23].

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
