# Peer review of "Properties of Therapeutic Deep Eutectic Solvents of l-Arginine and Ethambutol for Tuberculosis Treatment"

_molecules, 2018, doi:10.3390/molecules24010055_

Round 1
Reviewer 1 Report
The authors propose a new method to increase the bioavailability of anti-turberculosis drugs, L-arginine and ethambutol, by incorporating them in therapeutic deep eutectic solvents (THEDES). Based on the results in the manuscript, the strategy shows a great potential.
However, the main drawback is the discussion of characterization methods POM and DSC. POM measurements for the liquid phase at room temperature provided that the THEDES is homogenous without liquid crystals. If POM were done on the solid THEDES, more information could be obtained. If the objective of the POM is to just show that homogeneous liquids were obtained, the images could be transferred to supplementary information (discussion should be kept in the manuscript) since they are just black and do not provide additional information.
DSC thermograms show that water evaporates at 100 °C, with no phase transition through the whole temperature range. The lower limit of tested temperature range is 5 °C, which is quite high for a system considered as deep eutectic that contains water as a constituent with more than 70 mol%, i.e. melting temperature of water is 0 °C therefore the liquidus temperature should be around this temperature if not much lower. In DSC usually the lower and upper boundaries of temperature range should be at least 50 degrees from temperature of interest. As discussed in the paper, however, DSC results prove the temperature stability of THEDES.
Section 2.3: Was the water content measured in the samples that were just prepared by weight before? If yes, why is the deviation between the intended content so high? For example in CA:Ethambutol:H2O 2:1:10 the water content should be around 23.4 wt% calculated from the mole ratio, but only 12.4 % were measured (M(ethambutol)=204.1 g/mol; M(citric acid)=192.12 g/mol). Or the high deviation for CA:L-arginine:H2O 1:1:6 and 1:1:7 and 2:1:8 with expected 22.8, 25.6, and 20.52 wt% (M(L-arginine)=174.2 g/mol). Is there an explanation for this?
Lines 90-91: The second peak in the ethambutol thermogram might be also due to crystalline-crystalline transition and not only crystalline-amorphous transition. On the other hand, the peak in the L-Arginine thermogram is most probably water evaporation since it is around 100 C and it was confirmed that L-arginine has some water content.
Line 105: Aren’t the hermetic capsules called crucibles?
Line 109: the absence of any peak in the thermogram means that the mixture does not crystalize at temperatures above 5 C. However, this might be because of kinetic limitation. Many DES systems do not crystalize in DSC. The conclusion is strong and it is not supported well with an evidence.
Line 118: Is the water content listed in wt%? Please indicate the kind of percentage.
Line 234: DES are not synthesized, they are just mixtures. The challenge is because of the large number of compounds that can form eutectic mixtures. Interactions at molecular level does not affect preparation of DES, but it is more important to find the eutectic ratio between components to form a DES that is liquid at room temperature. Please reformulate the sentence.
Lines 243-245: “According to literature, the systems must have melting temperatures inferior to the melting peaks of the individual components [6,16], which was observed in our systems”
water is a component in THEDES and it melts at 0 °C. The mixtures were only tested above 5 °C.
Lines 245-246:” the formation mechanism of the deep eutectic solvents should not be restricted to simple eutectic melting point”
Why should it not be restricted to simple eutectic melting point? More explanation is required.
Lines 263-269: Is there a possible explanation why permeability and solubility were increased for ethambutol, but not for L-arginine? In this design of permeability measurement, will it always be expected that for low solubility in PBS (receptor compartment) the permeability will also be low?
Line 355: Polyethersulfone?
Author Response
Point 1: The authors propose a new method to increase the bioavailability of anti-tuberculosis drugs, L-arginine and ethambutol, by incorporating them in therapeutic deep eutectic solvents (THEDES). Based on the results in the manuscript, the strategy shows a great potential. However, the main drawback is the discussion of characterization methods POM and DSC. POM measurements for the liquid phase at room temperature provided that the THEDES is homogenous without liquid crystals. If POM is to just show that homogeneous liquids were obtained, the images could be transferred to supplementary information (discussion should be kept in the manuscript) since they are just black and do not provide additional information.
Response 1: The POM images demonstrate the homogeneous liquids obtained and are part of the characterization of the mixture, because of that they are presented in the main text as a proof of the formation of liquid eutectic mixtures.
Point 2: DSC thermograms show that water evaporates at 100ºC, with no phase transition through the whole temperature range. The lower limit of tested temperature range is 5ºC, which is quite high for a system considered as deep eutectic that contains water as a constituent with more than 70 mol%, i.e. melting temperature of water is 0ºC therefore the liquidus temperature should be around this temperature if not much lower. In DSC usually the lower and upper boundaries of temperature range should be at least 50 degrees from temperature of interest. As discussed in the paper, however, DSC results prove the temperature stability of THEDES.
Response 2: The authors did not check, in fact, the crystallization temperature of the DES. However, DSC results showed a temperature stability of THEDES in the range of temperature tested and we can observe the water evaporation, approximately at 100ºC. We consider that the range of temperatures tested is enough to see the stability of the compounds, in a temperature window of 5 ºC to 230 ºC (with the exception of ethambutol which was performed up to degradation temperature of the compounds ~260 ºC). As they are intended for medical usage the systems should be liquid at T~36 ºC. The results obtained further proof that the eutectic mixture is formed.
Point 3: Section 2.3: Was the water content measured in the samples that were just prepared by weight before? If yes, why is the deviation between the intended content so high? For example in CA:Ethambutol:H2O 2:1:10 the water content should be around 23.4 wt% calculated from the mole ratio, but only 12.4 % were measured (M(ethambutol)=204.1 g/mol; M(citric acid)=192.12 g/mol). Or the high deviation for CA:L-arginine:H2O 1:1:6 and 1:1:7 and 2:1:8 with expected 22.8, 25.6, and 20.52 wt% (M(L-arginine)=174.2 g/mol). Is there an explanation for this?
Response 3: The ratio of water was added when the eutectic mixture was prepared and then the water content of the mixtures was measured. The fact that the water content is lower than the expected by the calculations of molar ratio, probably is due to the fact that water is part of the eutectic mixture and interacts with the other components, decreasing the value of the water content present in the final mixture. Being this value the moisture content of the sample.
Point 4: Lines 90-91: The second peak in the ethambutol thermogram might be also due to crystalline-crystalline transition and not only crystalline-amorphous transition. On the other hand, the peak in the L-Arginine thermogram is most probably water evaporation since it is around 100 C and it was confirmed that L-arginine has some water content.
Response 4: Yes, the existence of different peaks in ethambutol thermograms might be related with different phase transitions. In respect, to L-arginine it is reported in the text that the peak around 100ºC it could be due to water evaporation. The authors have rephrased this in the manuscript to clarify the readers.
Point 5: Line 105: Aren’t the hermetic capsules called crucibles?
Response 5: General the capsules used in DSC are called capsules or pans. A crucible is a ceramic or metal container in which metals or other substances may be melted or subjected to very high temperatures, such as for example is the case in thermogravimetric analysis (TGA). In this work hermetic capsules were used.
Point 6: Line 109: the absence of any peak in the thermogram means that the mixture does not crystalize at temperatures above 5 C. However, this might be because of kinetic limitation. Many DES systems do not crystalize in DSC. The conclusion is strong, and it is not supported well with an evidence.
Response 6: The authors agree with the referee’s comment. The authors have performed stability studies at room temperature and have visually observed if there were any signs of crystallization. The systems have demonstrated to be very stable for up to 6 months.
Point 7: Line 118: Is the water content listed in wt%? Please indicate the kind of percentage.
Response 7: Yes, the water content is listed in wt%. This was corrected in the text.
Point 8: Line 234: DES are not synthesized, they are just mixtures. The challenge is because of the large number of compounds that can form eutectic mixtures. Interactions at molecular level does not affect preparation of DES, but it is more important to find the eutectic ratio between components to form a DES that is liquid at room temperature. Please reformulate the sentence.
Response 8: The authors have reformulated the text following the referee suggestion.
Point 9: Lines 243-245: “According to literature, the systems must have melting temperatures inferior to the melting peaks of the individual components [6,16], which was observed in our systems” water is a component in THEDES and it melts at 0 °C. The mixtures were only tested above 5 °C.
Response 9: The presence of water is fundamental for the preparation of a liquid system at room temperature. Evidences provided by NMR show that there are interactions between the three components of the mixture. So even though, the authors indeed did not perform DSC measurements bellow 5ºC it is expected that the water melting point decreases bellow 0ºC. the authors do not consider this as a major issue for the purpose of the manuscript herein presented. Furthermore, the range of temperatures tested showed stable mixtures without any melting peaks, until the water content is evaporated.
Point 10: Lines 245-246:” the formation mechanism of the deep eutectic solvents should not be restricted to simple eutectic melting point” Why should it not be restricted to simple eutectic melting point? More explanation is required.
Response 10: The formation mechanism of the deep eutectic solvents could be explained by the decrease of the melting point, however another low transition temperature mixtures could be characterized by this point, and because of that we need another type of characterization as NMR, for example, to confirm the existence of an eutectic mixture. By NMR we observe the possible intermolecular interactions between the components; POM is important too for the observation of homogeneous liquids without the formation of crystals.
Point 11: Lines 263-269: Is there a possible explanation why permeability and solubility were increased for ethambutol, but not for L-arginine? In this design of permeability measurement, will it always be expected that for low solubility in PBS (receptor compartment) the permeability will also be low?
Response 11: In case of solubility of L-arginine that is also high, so the increase is not so pronounced like in ethambutol. The permeability studies in L-arginine did not show an increase of permeability. The permeability measurement is not just conditioned by the solubility of the compound in the given solvent, but also with the interaction of the compound with the membrane, in this case PES-U (polyethersulfone) and the ability of the compound to permeate this membrane. The same compound in the same solvent could have low permeability and high solubility and vice-versa, like reported by Biopharmaceutics Classification System for several active pharmaceutical ingredients.
Point 12: Line 355: Polyethersulfone?
Response 12: The authors have corrected the mistake.

Reviewer 2 Report
Review (recommended minor revision)
Overall, sentences are too long; text will be more understandable, if they are shortened. Passive should be used correctly (time, singular/ plural, the position of subject…). The Materials and Methods chapter should be after the Introduction; it’d be better, if you combined Results/Discussion.
***attached review file***

Author Response
Point 1: Overall, the sentences are too long, it is more understandable if they are shortened. Passive should be used correctly (singular/plural, time, the position of subject). Materials and methods chapter should be after the Introduction, maybe it would also be better if you combine the chapters Results and Discussion. The resolution of Figures which include graphs should be higher. Otherwise, the text is written simply and clearly.
Response 1: The authors acknowledge the reviewer comments. The manuscript was revised and changes in the manuscript according to the indications of the reviewer were made. Particularly, the sentences were shortened, and some small mistakes were corrected. In relation to the structure of the manuscript it was not changed, because the structure is the one that the journal indicated.

Round 2
Reviewer 1 Report
Author’s response 3: The ratio of water was added when the eutectic mixture was prepared and then the water content of the mixtures was measured. The fact that the water content is lower than the expected by the calculations of molar ratio, probably is due to the fact that water is part of the eutectic mixture and interacts with the other components, decreasing the value of the water content present in the final mixture. Being this value the moisture content of the sample.
Reviewer’s answer: The explanation why the water content is deviating from the expected is not satisfactory. The authors should clarify what they mean by the interaction of water with other components decreases the value of water content and how they expect this to change the analytical results. For such high deviation a comparison with expected values and a discussion should be given in the manuscript. Please revise the definition of eutectic mixtures. There is no reaction except the eutectic reaction. In eutectic reaction, the liquid mixture is in equilibrium with two or more solid phases at eutectic temperature. There should not be a reaction where water is consumed and decreased in this mixture. It has been shown that the systems are liquid at room temperature. Therefore, the water content was tested above the eutectic temperature, i.e. no eutectic reaction. The measured water content must be the same or larger in case of L-Arginine since it is very hygroscopic and it is expected to increase the water content. The authors should take into consideration whether the problem is the measuring method. For such high water contents, Coulometer KF might not be applicable. In case a dilution was made, then the error might be due to dilution. Below quote was taken from the instruction manual of Metrohm 831 KF Coulometer:
“The water content of the working medium influences the stoichiometry of the KF reaction. If it exceeds 1 mol/L (18 g/L) then the reaction behavior changes to favor the Bunsen reaction for aqueous solutions. This means that 2 H2O are consumed for one I2 or one SO2. It is uneconomic to titrate such a high water content in a large volume of sample. In this case it is important to dilute the sample and/or to use a small amount of sample.”
It is not clearly described in the Methods Section how the samples were prepared. More details are required about the complete method in case of dilution. The measurements should be repeated and checked against another method.
Author’s response 10: The formation mechanism of the deep eutectic solvents could be explained by the decrease of the melting point, however another low transition temperature mixtures could be characterized by this point, and because of that we need another type of characterization as NMR, for example, to confirm the existence of an eutectic mixture. By NMR we observe the possible intermolecular interactions between the components; POM is important too for the observation of homogeneous liquids without the formation of crystals.
Reviewer’s answer: Eutectic mixtures are mixtures; the depression of melting temperature of the simple eutectic mixture depends on the solid phase of the pure compounds as well as the non-ideality in the liquid mixture. NMR is used to confirm the chemical shift of elements due to intermolecular interactions, and this corresponds to liquid phase non-ideality. The definition of simple eutectic is restricted to binary systems where the components have complete immiscibility in the solid phase. Therefore, simple eutectic mixtures do not correspond to “simple mixtures”, complex interactions in the liquid phase can exist as well for simple eutectic systems. The authors should revise Lines 245-246 in order to avoid confusion.
Author Response
Author’s response 3: The ratio of water was added when the eutectic mixture was prepared and then the water content of the mixtures was measured. The fact that the water content is lower than the expected by the calculations of molar ratio, probably is due to the fact that water is part of the eutectic mixture and interacts with the other components, decreasing the value of the water content present in the final mixture. Being this value the moisture content of the sample.
Reviewer’s answer: The explanation why the water content is deviating from the expected is not satisfactory. The authors should clarify what they mean by the interaction of water with other components decreases the value of water content and how they expect this to change the analytical results. For such high deviation a comparison with expected values and a discussion should be given in the manuscript. Please revise the definition of eutectic mixtures. There is no reaction except the eutectic reaction. In eutectic reaction, the liquid mixture is in equilibrium with two or more solid phases at eutectic temperature. There should not be a reaction where water is consumed and decreased in this mixture. It has been shown that the systems are liquid at room temperature. Therefore, the water content was tested above the eutectic temperature, i.e. no eutectic reaction. The measured water content must be the same or larger in case of L-Arginine since it is very hygroscopic and it is expected to increase the water content. The authors should take into consideration whether the problem is the measuring method. For such high water contents, Coulometer KF might not be applicable. In case a dilution was made, then the error might be due to dilution. Below quote was taken from the instruction manual of Metrohm 831 KF Coulometer:
“The water content of the working medium influences the stoichiometry of the KF reaction. If it exceeds 1 mol/L (18 g/L) then the reaction behavior changes to favor the Bunsen reaction for aqueous solutions. This means that 2 H2O are consumed for one I2 or one SO2. It is uneconomic to titrate such a high water content in a large volume of sample. In this case it is important to dilute the sample and/or to use a small amount of sample.”
It is not clearly described in the Methods Section how the samples were prepared. More details are required about the complete method in case of dilution. The measurements should be repeated and checked against another method.
Author’s response: The authors have carefully revised this and have also read the manual of the Karl Fisher equipment and agree with the refere that perhaps this is not the most suitable method for the determination of the water content of the samples. As the results might be missleading the authors have decided to withdraw this information from the manuscript. Water content is then only given as a molar ratio of the components present.
Author’s response 10: The formation mechanism of the deep eutectic solvents could be explained by the decrease of the melting point, however another low transition temperature mixtures could be characterized by this point, and because of that we need another type of characterization as NMR, for example, to confirm the existence of an eutectic mixture. By NMR we observe the possible intermolecular interactions between the components; POM is important too for the observation of homogeneous liquids without the formation of crystals.
Reviewer’s answer: Eutectic mixtures are mixtures; the depression of melting temperature of the simple eutectic mixture depends on the solid phase of the pure compounds as well as the non-ideality in the liquid mixture. NMR is used to confirm the chemical shift of elements due to intermolecular interactions, and this corresponds to liquid phase non-ideality. The definition of simple eutectic is restricted to binary systems where the components have complete immiscibility in the solid phase. Therefore, simple eutectic mixtures do not correspond to “simple mixtures”, complex interactions in the liquid phase can exist as well for simple eutectic systems. The authors should revise Lines 245-246 in order to avoid confusion.
Author’s response: The authors have revised the sentence according to the reviewer suggestion. Line 244-247 “According to literature, the systems must have melting temperatures inferior to the melting peaks of the individual components [6,16]. In binary mixtures these observations lead to the absence of melting peaks of each individual component in DSC diagram [19,21].”
